# The Genetic Mechanism of the Immune Response to the Rice False Smut (RFS) Fungus *Ustilaginoidea virens*

**DOI:** 10.3390/plants12040741

**Published:** 2023-02-07

**Authors:** Dewei Yang, Niqing He, Fenghuang Huang, Yidan Jin, Shengping Li

**Affiliations:** 1Institute of Rice, Fujian Academy of Agricultural Sciences, Fuzhou 350018, China; 2College of Agriculture, Fujian Agriculture and Forestry University, Fuzhou 350002, China; 3Plant Immunity Center, Fujian Agriculture and Forestry University, Fuzhou 350002, China

**Keywords:** rice, rice false smut, research progress, genetic mechanisms, QTL, resistance protein

## Abstract

Rice false smut (RFS), which is caused by *Ustilaginoidea virens* (*U. virens*), has become one of the most devastating diseases in rice-growing regions worldwide. The disease results in a significant yield loss and poses health threats to humans and animals due to producing mycotoxins. In this review, we update the understanding of the symptoms and resistance genes of RFS, as well as the genomics and effectors in *U. virens*. We also highlight the genetic mechanism of the immune response to RFS. Finally, we analyse and explore the identification method for RFS, breeding for resistance against the disease, and interactions between the effector proteins and resistance (R) proteins, which would be involved in the development of rice disease resistance materials for breeding programmes.

## 1. Introduction

*Ustilaginoidea virens* (*U. virens*) Takahashi (teleomorph: *Villosiclava virens*), a pathogenic ascomycota fungus, causes a destructive grain disease in rice that is known as rice false smut (RFS). RFS was first reported in 1878 in the Tirunelveli district of Tamil Nadu, India [1]. RFS has long been classified as a mild disease in rice production, occurring sporadically in some rice-growing regions, such as South and East Asia. However, it has recently become one of the most important diseases after rice blast and sheath blight in most rice-growing regions of the world, and it has attracted increasing attention from researchers and farmers. In China, the average annual incidence area of RFS is 3.06 million hectares, resulting in an annual yield loss of 158.6 million kg. From 2008 to 2016, the average annual control area of RFS was 6.92 million hectares [2]. The incidence of the disease is higher, especially in the rice-growing areas of the middle and lower reaches of the Yangtze River [2]. In India, the proportion of tillers infected with false smut balls has been reported to be between 5% and 85%, with the disease causing a yield loss of 0.2–49%, depending on the rice variety and disease intensity [3,4]. Outbreaks of RFS have been reported in other regions, including the Middle East and North America [5].

Understanding the life cycle of *U. virens* is of great significance for prevention, control and study of its pathogenesis [6]. As a biotroph fungus, *U. virens* can be cultured under laboratory conditions, and it relies on a live host for both asexual and sexual reproduction [6]. On the other hand, *U. virens* can reproduce sexually in both field and laboratory conditions [7,8,9,10]. *U. virens* ascospores and chlamydospores can infect rice spikelets [11,12,13]. After successful infection of rice flowers, *U. virens* transforms into false smut balls covered with powdery chlamydospores [14]. These false smut balls range in colour from yellow-orange to bluish-black and often produce sclerotia when exposed to large diurnal temperature differences [15]. In addition, *U. virens* ascospores and chlamydospores may contaminate rice seeds and/or be spread by air currents and rain splashes in rice fields where they survive the winter [16,17]. Under the condition of high humidity and more water, some chlamydospores germinate to form mycelia and produce a large number of secondary conidia, which can re-infect rice flowers in the late booting stage [6]. However, it is still difficult to accurately track and identify the germination of overwintered chlamydospores in rice fields [6]. The formation of false smut balls not only hinders grain filling but also increases the sterility of the spikelets adjacent to the balls, resulting in considerable yield loss [18]. Most importantly, false smut balls produce toxic mycotoxins that threaten human and animal health [19,20].

During fungal invasion, rice cells first produce a nonspecific immune response, which is called pathogen-associated molecular pattern (PAMP)-triggered immunity (PTI), after recognizing the pathogen-associated molecular patterns (PAMPs) of pathogenic fungus, such as chitin. After interacting with receptors, pathogenic bacteria secrete a large number of effector proteins to neutralize PTI in sensitive rice plants. Certain fungal effectors are recognised by homologous nucleotide-binding leucine-rich repeat (NLR) proteins, leading to a powerful immune response in resistant rice plants that is known as effector-triggered immunity (ETI) [21,22,23,24,25,26]. PTI and ETI trigger a number of similar immune responses, including mitogen-activated protein kinase activation, transient calcium influx, rapid bursts of reactive oxygen species (ROS), callose deposition, transcriptional reprogramming, and plant hormone regulation [27,28]. Although it is traditionally believed that plant ETI and PTI respond independently, they are not invariably seen in isolation. Recent studies have shown that PTI and ETI are closely related and enhance each other, providing strong disease resistance. Plant PTI is an indispensable part of ETI in the process of pathogen infection and, in turn, the activation of ETI can enhance the immune response stimulated by PTI [29,30]. This paper reviews the phenotypic symptoms of RFS, the identification of resistance (R) genes, and the genome and effector proteins of *U. virens*, and it focuses on the molecular mechanism of the rice immune response to RFS. The method for resistance identification, the breeding of resistance, and the interaction between the R protein and effector protein are also discussed in depth to provide a reference for research on RFS.

## 2. Genomics and Effectors in *Ustilaginoidea virens*

A new era for *U. virens* functional studies began with the availability of genome sequences in 2014 [31]. Currently, genome sequences of multiple *U. virens* isolates, such as *UV*-8b, IPU010 and *UV*-GVT from China, Japan and India, respectively, are available in the NCBI database [32]. The assembled genome sizes of these isolates vary from 26.97 to 39.40 Mb, encoding 6451 to 8426 predicted proteins. The difference in genome size is largely due to repetitive elements. Although the *U. virens* genome possesses fewer genes associated with cell wall degradation and nutrient uptake, it is rich in the genes involved in pathogen–host interactions, which account for 13.1% of the total protein-coding genes [31]. Importantly, the *U. virens* genome and the predicted genome-wide protein interactome network remarkably accelerate the functional identification of pathogenicity genes and virulence effectors [33].

Effectors are an important set of pathogenic agents that can disable plant immunity and/or manipulate cellular processes to promote successful colonisation of invasive organisms. Some of the identified effectors consist of more than 400 amino acids and fewer than 4 cysteine (Cys) residues [34,35] (Table 1). Analysis has shown that 193 of the 628 secreted proteins were predicted to be candidate effector proteins for *UV*-8b based on the fact that they have signalling peptides, lack transmembrane domains, and contain fewer than 400 amino acids and more than 4 Cys residues [31]. A total of 256 candidate effector proteins with signal peptides and 165 atypical effector proteins were recently predicted in the *U. virens* genome, which are usually short peptides enriched in Cys residues and may be located in the exoplasmic space of host cells [36].

Through the screening of immunosuppressive effector proteins in *U. virens*, UV_1261/SCRE2, UV_5215, and UV_2964 were determined to inhibit *Burkholderia glumae*-triggered cell death to different degrees in *Nicotiana benthamiana* [31]. The effectors SCRE1 and UV_1261/SCRE2 inhibited highly sensitive cell death induced by the mouse proapoptotic protein Bax and *Phytophthora infestans* elicitin INF1, and they also inhibited immunity induced by pathogen-associated molecular patterns in rice [37,38]. Recent studies showed that the expression of the effector SCRE4 repressed the transcription of the immunopositive regulatory factor OsARF17 in *U. virens*, which promoted infection [39]. In addition, the effector SCRE6, a phosphatase of *Aspergillus oryzae*, interacted with rice mitogen-activated protein kinase 6 (OsMPK6) to dephosphorylate OsMPK6 and enhance its stability, thereby negatively regulating the rice immune response [40]. One study found that UvHrip1 suppressed the cell death symptoms and ROS accumulation in *Nicotiana benthamiana* triggered by *Burkholderia glumae* [41]. However, how these effectors affect the infection process of rice flowers has not been fully determined [42,43]. Therefore, the next most important and urgent task is to determine the host targets of *U. virens* effectors, which could serve as excellent probes to identify unknown components of plant immunity and metabolism.

## 3. Symptoms and Resistance-Related Genes of RFS

The fungus forms black, dark green, and yellow false smut balls in the rice panicle. At the early stage of the false smut balls, the spots are small and confined to the glumes of the rice inflorescence. The typical symptoms of false smut balls are yellow or dark green smut balls in the middle and lower panicle of the rice (Figure 1). The surface of false smut balls is powdery and cracked. Later, it gradually expands and, finally, forms a complete rice ball with a diameter of more than 1 cm, which can completely wrap the whole inflorescence. Generally, a diseased kernel contains sclerotia, and some contain 2–4 sclerotia. However, there is a lack of systematic research on the genes related to false smut ball development and the associated formation mechanism [13].

In recent years, researchers have carried out much research on the determination of the resistance genes of RFS and identified some related QTLs (Table 2). For example, Li et al. constructed 157 recombinant inbred lines using resistance material IR28 as a parent and identified 7 QTL loci related to RFS, namely *qFsrl*, *qFsr2*, *qFsr4*, *qFsr8*, *qFsr10*, *qFsr11* and *qFsr12* [44]. Xu et al. identified two QTLs related to RFS, namely *QFsr10* and *QFsr12*, using near-isogenic gene lines [45]. Yuan et al. used MR183-2 and 08R2394 as research materials to identify four QTLs related to RFS resistance, namely *qFsr2-1*, *qFsr2-2*, *qFsr3-1* and *qFsr8-1* [46]. Zhou et al. identified four QTLs related to RFS, namely *qFSR-6-7*, *qFSR-10-5*, *qFSR-10-2* and *qFSR-11-2*, using 213 infiltration lines of Lemont/Teqing constructed as research materials [47]. Qiu et al. used Nanjing 11 as the research material and mapped the RFS-related gene *FSR1* to the region of 220 kb on chromosome 1 of rice through mapping cloning technology and preliminarily concluded that LOC_Os01g42630 was its candidate gene [48]. Long et al. identified LOC_Os01g15580 on chromosome 1 (12.3 kb) as a candidate gene for resistance to RFS through a genome-wide association analysis [49]. Recently, Neelam et al. used RYT2668 as research material and mapped seven QTLs related to RFS resistance: *qRFSr5.3*, *qRFSr7.1a*, *qRFsr9.1*, *qRFSr2.2*, *qRFSr4.3*, *qRFSr5.4* and *qRFSr7.1b* [50]. In addition, Huang et al. identified three QTLs (*qFSR10*, *qFSR2* and *qFSR9*) related to RFS resistance in the Japonica rice varieties XS47 and XS664, which were located on chromosomes 2, 10 and 11, respectively, and finely mapped *qFSR2* among them. This study provided a foundation for the subsequent cloning of the resistance genes of rice smut disease and resistance breeding [51].

However, no QTL for resistance to RFS has been cloned thus far. Moreover, the identification, screening and genetic research of rice germplasm resources for resistance to RFS have not been carried out in depth. In addition, it remains to be studied whether there are genes for resistance to smut in rice and whether differences in panicle traits (such as upright panicle, loose panicle, dense panicle, large panicle, long panicle, etc.) have an impact on RFS resistance. We studied the spreading panicle mutant (*spr9*) of rice and found that the *spr9* mutant showed more resistance to rice smut than the wild type (He, N.Q. & Yang, D.W., unpublished data).

## 4. Genetic Mechanism of the Immune Response to RFS

At present, although many effector proteins have been identified in *U. virens*, no R gene of RFS has been cloned. Therefore, no reliable model of the ETI immune reaction between the R and effectors has been established in RFS.

For the immune response formed by *U. virens* and rice, researchers initially formed a model of the PTI immune response triggered in rice through a series of related studies (Figure 2a). Song et al. [52] identified MAMP SGP1 conserved in fungi from *U. virens* secretions. SGP1 acts on the extracellular bodies of plant cells and depends on BAK1, the core receptor of plant immunity [53]. SGP1 encodes a serine and threonine sugar-based phosphatidyl inositol glycosylphosphatidylinositol (GPI)-anchored protein, which is ubiquitous in fungi. Plants specifically recognise 22 conserved amino acids in SGP1, and this peptide of the 22 amino acids (SNP22) is sufficient to induce cell death and disease resistance in plants [52]. Recent studies have shown that SCRE6, a phosphatase effector of SMPK6, is secreted and transferred to rice cells during infection and then interacts with mitogen-activated protein kinase 6 (OsMPK6) and dephosphorylates, which negatively regulates the immune response of rice. Further studies showed that the dephosphorylation of OsMPK6 enhanced its stability and inhibited the immunity of rice [40]. Chen et al. [54] showed that the toxic effect protein UvSec117 of *U. virens* enters the host cell, interacts with the rice histone deacetylase OsHDA701, and recruits a large amount of OsHDA701 into the nucleus through the nuclear localisation signal of UvSec117. At the same time, the interaction between UvSec117 and OsHDA701 can enhance the deacetylation activity of OsHDA701, reduce the acetylation level of rice histone from two aspects, and then interfere with the activation of defence genes and host immunity. Li et al. [55] found that the cytoplasmic effector UvCBP1 secreted by *U. virens* interacts with the rice scaffold protein OsRACK1A to compete for its interaction with NADPH oxidase OsRBOHB, resulting in the inhibition of reactive oxygen species (ROS) production. Further studies showed that overproduction of OsRACK1A could restore the OsRACK1A–OsRBOHB association and promote the phosphorylation of OsRBOHB to increase ROS production, resulting in resistance to *U. virens* without affecting the rice yield. The researchers found that the *Arabidopsis* homologous protein RACK1 was involved in the MAPK signalling pathway [56], which is consistent with our speculation that OsRACK1A may be involved in the MAPKKK signalling pathway (Figure 2a).

Another model is the immune response mediated by the SnRK1A-XB24 phosphorylation cascade signalling pathway (Figure 2b). Yang et al. [57] found that SnRK1A, a conserved energy-regulating protein in eukaryotes, interacts with XB24 and phosphorylates its Thr83 amino acid to improve its ATPase activity and regulate the immune signalling pathway of rice. During the early stage of infection, SCRE1 specifically interacts with XB24 in rice, blocking the binding of XB24 to ATP and inhibiting its hydrolysis of ATP. SCRE1, on the other hand, competitively inhibits the interaction of SnKR1A with XB24 and, thus, inhibits the SnRK1A-XB24 phosphorylation cascade and ATPase activity.

## 5. Discussion

### 5.1. Methods for the Identification of Resistance to RFS

Artificial inoculation is an effective method for identifying the resistance of different rice varieties to RFS. It is mainly divided into injection inoculation, which sprays and spreads the false smut balls directly into the rice field, and it has achieved good results in resistance identification. However, manual injection is time-consuming, labour-intensive and inefficient. Spray inoculation combined with natural identification was shown to be beneficial in terms of increasing the incidence and efficiency of rice false smudges. Researchers have conducted many studies on the classification standard of RFS resistance, but there is no unified classification standard at present. The grading criteria of the International Rice Research Institute (IRRI) can not only better reflect the severity of the disease but also be convenient for field operations, although there are some limitations in production practice. For example, the evaluation results of the same rice variety or strain with different sizes are different according to this grading standard. We inoculated Zhonghua 11 with *U. virens* and found that the large grain had 5–9 different diseased grains, while the small grain had 3–5 different diseased grains. In addition, the number of grains per panicle of conventional Indica and Japonica rice is approximately 200, while the number of grains per panicle of some super rice varieties is more than 300, with some even reaching more than 350. It has also been proposed to evaluate the grade of the disease, which is calculated using the average number of false smut balls per ear, namely the number of diseased grains per ear [58]. Obviously, if the disease grade is calculated according to the number of diseased grains per panicle, under the condition of the same number of diseased grains per panicle (that is, the disease grade is the same), the varieties with more grains per panicle will have a very low disease grain rate, while the varieties with fewer grains per panicle will have a very high disease grain rate. Obviously, if the grade is divided according to the IRRI grading standards, the results are obviously different. Therefore, only by establishing a unified classification standard for the resistance identification of RFS can a good foundation be provided for the resistance evaluation of rice smut germplasm resources and the mining of resistance genes [59,60,61].

### 5.2. Identification of Resistance Genes and Molecular Breeding for RFS

By using different genetic populations, researchers identified some QTLs related to rice smut resistance (Table 2). However, no major resistance genes for resistance to RFS were found by mapping cloning, and it was speculated that the resistance mainly came from quantitative trait loci. Moreover, the identification, screening and genetic research of rice germplasm resources for resistance to RFS have not been carried out in depth by researchers. It is speculated that the selection and identification of resistant varieties are relatively delayed due to the difficulty of artificial inoculation and the lack of screening conditions for resistance to smut. Furthermore, whether the differences in panicle traits (such as upright panicle, loose panicle, dense panicle, large panicle, long panicle, etc.) affect the resistance of RFS remains to be further studied. Finally, the depth and breadth of the identification and screening of rice germplasm resources resistant to RFS have not been sufficient, the utilisation efficiency of the existing rice germplasm resistance is not high, and the breeding of rice hybrid varieties resistant to rice smut should be strengthened.

Therefore, on the basis of establishing a standard and unified identification technology for resistance to RFS, field screening of resistant rice varieties should be strengthened and resistant germplasm resources should be explored. By exploiting and utilising the resistance sources and variants of existing germplasm resources, the resistance genes of RFS were cloned, and those resistance genes were introduced into varieties with excellent comprehensive traits (such as yield and quality traits, etc.). By means of molecular breeding, more high-yield, high-quality and disease-resistant rice varieties were cultivated and then popularised in production practices.

In recent years, the host-induced gene silencing (HIGS) technique has been used to silence fungus-specific pathogenic factors in *U. virens*, which led to the successful development of new rice materials with a high resistance to RFS. By using HIGS, Li et al. [62] obtained a novel rice material resistant to RFS by inhibiting the expression of two key chitin synthesis genes (*UvChs2* and *UvChs5*). Chen et al. [63] identified a new septin gene, *UvAspE*, in *U. virens* and found that *UvAspE* was involved in the regulation of hyphal growth, septum development and pathogenicity of *U. virens* through gene knockout and other means. Then, RNAi (*UvAspE*, *UvCom1* and *UvPro1*) transgenic rice lines were constructed and artificial inoculation with rice smut was carried out in the laboratory for identification purposes. It was proven that the created transgenic rice could significantly improve resistance to RFS. These studies have demonstrated the feasibility and effectiveness of HIGS in rice resistance to RFS, and the results will provide new ideas and strategies for cultivating RFS-resistant materials and solving the bottleneck problem of the lack of RFS-resistant resources in rice production.

### 5.3. Mechanism of Interaction between Rice R Proteins and Effector Proteins in RFS

In plants represented by *Arabidopsis thaliana*, the study of the interaction between the pathogen and host has been basically established, including the mechanism of the interaction between the R proteins and effector proteins. There are two main interaction modes between the R proteins and effector proteins, namely the direct interaction mode and indirect interaction mode [64]. Indirect mutual modes mainly include the guard model [65,66], decoy model [67], integrated decoy model [68] and model of effector proteins regulating resistance R proteins [69,70].

Although many effector proteins have been identified (Table 1), the functions of some effector proteins have not been fully defined [71]. Importantly, the *R* gene for resistance to RFS has not been isolated. The mechanism of recognition and signal transduction between the R proteins and effector proteins and the genetic mechanism of the immune response remain unclear. Therefore, to solve these problems, researchers need to clone the *R* gene related to RFS, search for the corresponding effector protein, and further study the interaction between them to reveal the genetic mechanism of rice smut resistance.

## 6. Conclusions and Future Perspectives

*U. virens* has emerged as an important plant pathogenic fungi causing a serious rice-grain disease in recent years [36]. In this paper, we update the understanding of the symptoms and resistance genes of RFS, as well as the genomics and effectors in *U. virens*. At the same time, we preliminarily develop the genetic mechanism of the immune response to RFS. Finally, we analyse and explore the identification method for RFS, breeding for resistance to disease, and interactions between the effector proteins and R proteins. This study will provide a reference for the genetic mechanism of the immune response to RFS and breeding for disease resistance.

Despite these advances, important questions still need to be addressed in order to better understand the interaction between *U. virens* and rice. Is there a resistance gene in rice that can produce an immune response to *U. virens*? Is there a unique interaction pattern between *U. virens* and rice? How does the germination of chlamydospores occur after overwintering in rice fields? Why is the expression of ROS-related genes upregulated in rice induced by *U. virens* infection? In addition, mycoviruses infecting *U. virens* are widespread [72,73,74], and the change in the virulence of the fungus *U. virens* is also an interesting question. Overall, the *U. virens*–rice interaction is an interesting and specific pathological system, and any advances made will enrich our understanding of host–pathogen interactions.

## Figures and Tables

**Figure 1 plants-12-00741-f001:**
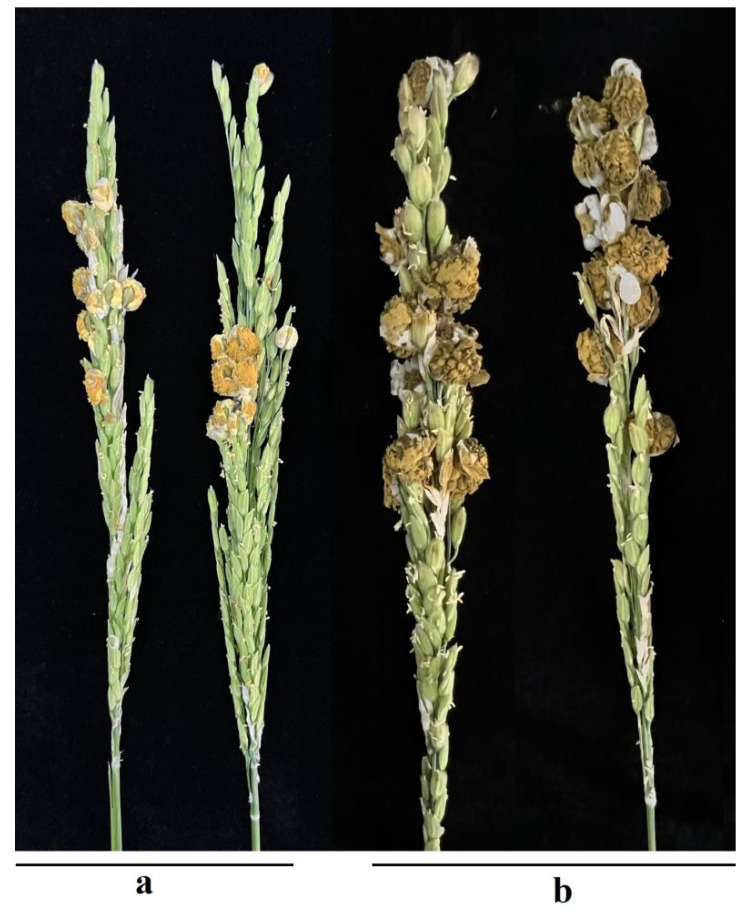
Symptoms of RFS. (**a**) A diseased rice panicle showing yellow false smut balls in a paddy field. (**b**) A diseased rice panicle showing greenish-black false smut balls in a paddy field.

**Figure 2 plants-12-00741-f002:**
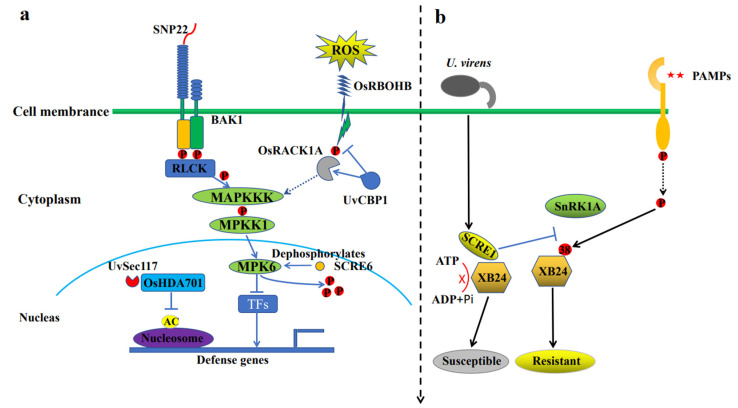
Model of the PAMP triggering the host immune response. (**a**) (I) Pattern recognition receptors (PRR) composed of some leucine-rich repeating (LRR-RLK) protein kinases and brassinosteroid receptor-associated kinase 1 (BAK1) in rice can detect the effector protein SNP22 in rice, thereby activating the downstream defence response; (II) after transfer, the phosphatase effector SCRE6 of *Aspergilus oryzae* interacts with the mitogen-activated protein kinase 6 (OsMPK6) to dephosphorylate OsMPK6 and enhance its stability, thus negatively regulating the rice immune response; (III) the toxic protein UvSec117 interacts with OsHDA701 and recruits a large amount of OsHDA701 to the nucleus, thereby enhancing the deacetylation activity of OsHDA701 and inhibiting chromosomal opening, which inhibits the binding of transcription-related proteins, thus interfering with the activation of defence genes and host immune response; and (IV) the effecting factor UvCBP1 interacts with the rice scaffold protein OsRACK1A and competes for its interaction with NADPH oxidase OsRBOHB, leading to the production of reactive oxygen species (ROS), which interferes with the defence host immune response. (**b**) The energy regulatory protein SnRK1A in rice interacts with XB24 and phosphorylates its amino acid Thr83 to increase the activity of its ATPase, activating the rice immune signalling pathway and showing resistance to disease. However, SCRE1, an effector secreted by SMF, specifically interacts with XB24 in rice, blocking the binding of XB24 to ATP. At the same time, SCRE1 competitively inhibits the interaction between SnKR1A and XB24, thereby inhibiting the phosphorylation cascade of SnRK1A-XB24 and the activity of ATPase, thus inhibiting the immune response of rice and showing susceptibility to the disease.

**Table 1 plants-12-00741-t001:** Effectors characterised in *Ustilaginoidea virens*.

Gene	Gene Product	Pathogenicity of Deletion Mutant	Gene Function	Reference
UV_1261/SCRE2	Effector protein	Reduced virulence	Suppresses cell death and pattern-triggered immunity in plant	[31,37]
UV_5215	Effector protein	Reduced virulence	Suppresses *B. glumae*-induced HR in *N. benthamiana*	[31]
UV_2964	Effector protein	Reduced virulence	Suppresses *B. glumae*-induced HR in *N. benthamiana*	[31]
SCRE1	Effector protein	Reduced virulence	Inhibits host immunity	[38]
SCRE4	Effector protein	Reduced virulence	An essential virulence effector and suppresses the expression of *OsARF17*	[39]
SCRE6	Effector protein	Reduced virulence	Interacts with and dephosphorylates the negative immune regulator OsMPK6 in rice, and suppresses plant immunity.	[40]
UvHrip1	Effector protein	Reduced virulence	Suppresses plant innate immunity and promotes disease development	[41]

**Table 2 plants-12-00741-t002:** QTLs associated with RFS.

Name	Chromosome	Resistant Material	Reference
*qFsrl*	1	IR28	[44]
*qFsr2*	2	IR28	[44]
*qFsr4*	4	IR28	[44]
*qFsr8*	8	IR28	[44]
*qFsr10*	10	IR28	[44]
*qFsr11*	11	IR28	[44]
*qFsr12*	12	IR28	[44]
*QFsr10*	10	Near-isogenic line	[45]
*QFsr12*	12	Near-isogenic line	[45]
*qFsr2-1*	2	MR183-2, 08R2394	[46]
*qFsr2-2*	2	MR183-2, 08R2394	[46]
*qFsr3-1*	3	MR183-2, 08R2394	[46]
*qFsr8-1*	8	MR183-2, 08R2394	[46]
*qFSR-6-7*	6	Near-isogenic line	[47]
*qFSR-10-5*	10	Near-isogenic line	[47]
*qFSR-10-2*	10	Near-isogenic line	[47]
*qFSR-11-2*	11	Near-isogenic line	[47]
*FSR1*	11	NanJing11	[48]
*LOC_Os01g15580*	1	315 core rice materials	[49]
*qRFSr5.3*	5	RYT2668	[50]
*qRFSr7.1a*	7	RYT2668	[50]
*qRFsr9.1*	9	RYT2668	[50]
*qRFSr2.2*	2	RYT2668	[50]
*qRFSr4.3*	4	RYT2668	[50]
*qRFSr5.4*	5	RYT2668	[50]
*qRFSr7.1b*	7	RYT2668	[50]
*qFSR10*	10	XS47, XS664	[51]
*qFSR2*	2	XS47, XS664	[51]
*qFSR9*	9	XS47, XS664	[51]

## Data Availability

All data are presented in the main text.

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
