# Peer review of "The Genetic Mechanism of the Immune Response to the Rice False Smut (RFS) Fungus Ustilaginoidea virens"

_plants, 2023, doi:10.3390/plants12040741_

Round 1

Reviewer 1 Report

My comments can be found in the attached MS.

Author Response

1、Did you mean fungus?

Response: Thank you for your careful review. Yes, We have changed “pathogenic bacteria” to “Pathogenic fungus”.

2、Please provide a brief discription of disease cycle.

Response: Thanks for the suggestion. We have added a brief discription of disease cycle.

3、Repeated word?

Response: Thank you for your careful review. Yes, and we have removed this duplicate.

4、Italicize the entire word.

Response: Thank you for your good advice. We have changed it to italics.

5、So initial symptoms are yellow in color?

Response: Normally, yellow is formed first, and then black or green black, but in some cases, black or green black is formed directly.

6、With what?

Response: Thank you for your careful review. In order to make it clear, we have modified the sentence "and then interacts with and dephosphorylates mitogen-activated protein kinase 6 (OsMPK6)" to" and then interacts with mitogen activated protein kinase 6 (OsMPK6) and dephosphorylates".

7、Italicize.

Response: Thank you for your good advice. We have changed it to italics.

8、effector?

Response: Thank you for your careful review. Yes, We have changed “effect” to “effector”.

9、Is this your own figure? If not please provide a reference.

Response: Yes. Our model is based on the results of previous studies and accumulated by our research group.

10、Please label these in the figure above.

Response: Thank you very much for your careful review. We've labeled these in the figure above.

11、Did you mean direct sowing of grain that was infested with U. virens?

Response: Thanks for the suggestion. Maybe we didn't describe it clearly enough. We have modified the sentence "direct sowing of Ustilaginoidea virens" to"spread the false smut balls directly into the rice field".

12、Did you mean field assessment?

Response: Yes, what we mean is that this evaluation method is convenient for field operation and evaluation

13、What do you mean by grade? I would expect either severity or incidence.

Response: We refer to the assessment of disease grade by the number of false smut balls in single-panicle rice. For clarity, we added this sentence: "It has also been proposed to evaluate the grade of disease using the number of false smut balls per ear of rice, which is calculated using the average number of false smut balls per ear of rice[48]."

In addition, we reviewed the manuscript again and made modifications for minor problems. All the modified parts of the paper were in revision mode.

Reviewer 2 Report

Please see the minor comments on the PDF file.

Author Response

1、authores should improve this information to avoid overlapping of information

Response: This is a very good point. In order to avoid information overlap, we have modified this part.

2、Conclusion? future perpective?

Response: Thank you for your good advice. In this paper, we have added conclusions and future perpective.

In addition, we reviewed the manuscript again and made modifications for minor problems. All the modified parts of the paper were in revision mode.

Round 2

Reviewer 1 Report

My comments can be found in the attached MS.

Author Response

Reviewer 1

1、It is not clear whether the fungus is biotroph or semi-biotroph. Could you be more specific?

Response: This is a very good point. U. virens is biotroph, in order to make this sentence more clear and explicit, we have changed the sentence "As a fungus U. virens can be cultured under laboratory conditions and is dependent on a live host for both asexual and sexual reproduction" to " As a biotroph fungus, U. virens can be cultured under laboratory conditions and relies on a live host for both asexual and sexual reproduction" .

2、Could you please change the wording here; "difficult to observe" sounds like problem related to eyes.

Response: Thank you for your good advice. We have changed the sentence "it is still difficult to observe the germination of overwintered chlamydospores in rice field" to " it is still difficult to accurately track and identify the germination of overwintered chlamydospores in rice field".

3、in China? From your introduction, it seems like an established pathogen in other countries like India.

Response: Thank you for your careful review. In recent years, the disease has occurred in China and other countries, including India. In 1878, samples of RFS collected from India were isolated and named the pathogen as Ustilaginoidea virens (U. virens). In order to be more accurate, we have changed the sentence "U. Virens has emerged as an important plant pathogen," to " U. Virens has emerged as an important plant pathogenic fungi".

4、Did you develop the genetic mechanism? Or your research identified the genetic mechanism?

Response: Thank you. We develop the genetic mechanism.

5、occurs?

Response: Thank you for your careful review. How do the germination of chlamydospores after overwintering in rice fields? We refer to how chlamydospores overwintered, spspores sprouted, and then infected rice again.

In addition, we reviewed the manuscript again and made modifications for minor problems. All the modified parts of the paper were in revision mode.